METHODS

# TransferGWAS of T1-weighted brain MRI data from UK Biobank

**Alexander Rakowski**[1], **Remo Monti**[1,2], **Christoph Lippert**[1,3]*

**1** Digital Health Machine Learning, Hasso Plattner Institute for Digital Engineering, University of Potsdam, Germany, **2** Max-Delbrück-Center for Molecular Medicine in the Helmholtz Association, Berlin Institute for Medical Systems Biology, Berlin, Germany, **3** Hasso Plattner Institute for Digital Health at Mount Sinai, New York, New York, United States of America

* christoph.lippert@hpi.de

**Data Availability Statement:** The PheWAS summary statistics for the phenotypes and polygenic scores, as well as the genetic correlations are available directly as supplementary material. The GWAS summary statistics as well as the weights of the fitted polygenic scores are

## Abstract

Genome-wide association studies (GWAS) traditionally analyze single traits, e.g., disease diagnoses or biomarkers. Nowadays, large-scale cohorts such as UK Biobank (UKB) collect imaging data with sample sizes large enough to perform genetic association testing. Typical approaches to GWAS on high-dimensional modalities extract predefined features from the data, e.g., volumes of regions of interest. This limits the scope of such studies to predefined traits and can ignore novel patterns present in the data. TransferGWAS employs deep neural networks (DNNs) to extract low-dimensional representations of imaging data for GWAS, eliminating the need for predefined biomarkers. Here, we apply transferGWAS on brain MRI data from UKB. We encoded 36, 311 T1-weighted brain magnetic resonance imaging (MRI) scans using DNN models trained on MRI scans from the Alzheimer's Disease Neuroimaging Initiative, and on natural images from the ImageNet dataset, and performed a multivariate GWAS on the resulting features. We identified 289 independent loci, associated among others with bone density, brain, or cardiovascular traits, and 11 regions having no previously reported associations. We fitted polygenic scores (PGS) of the deep features, which improved predictions of bone mineral density and several other traits in a multi-PGS setting, and computed genetic correlations with selected phenotypes, which pointed to novel links between diffusion MRI traits and type 2 diabetes. Overall, our findings provided evidence that features learned with DNN models can uncover additional heritable variability in the human brain beyond the predefined measures, and link them to a range of non-brain phenotypes.

## Author summary

Genome-wide association studies are a popular framework for identifying regions in the genome influencing a trait of interest. At the same time, the growing sample sizes of medical imaging datasets allow for their incorporation into such studies. However, due to high dimensionalities of imaging modalities, association testing cannot be performed directly on the raw data. Instead, one would extract a set of measurements from the images, typically using predefined algorithms, which has several drawbacks—it requires specialized

available at figshare: https://doi.org/10.6084/m9.figshare.25933717.v1 https://doi.org/10.6084/m9.figshare.25933663.v1.

**Funding:** This work was supported by the European Commission (Grant agreement ID: 101016775 to CL), the Deutsche Forschungsgemeinschaft (DFG, German Research Foundation) (via the research unit KI-FOR 5363 – 459422098 to CL), and the HPI Research School on Data Science and Engineering to AR.AR and RM received a salary from the European Commission (Grant agreement ID: 101016775). AR received a scholarship from the HPI Research School on Data Science and Engineering. The funders had no role in study design, data collection and analysis, decision to publish, or preparation of the manuscript.

**Competing interests:** The authors have declared that no competing interests exist.

software, which might not be available for new or less popular modalities, and can ignore features in the data, if they have not yet been defined. An alternative approach is to extract the features using pretrained deep neural network models, which are well suited for complex high-dimensional data and have the potential to uncover patterns not easily discoverable by manual human analysis. Here, we extracted deep feature representations of brain MRI scans from UK Biobank, and performed a genome-wide association study on them. Besides identifying genetic regions with previously reported associations with brain phenotypes, we found novel regions, as well as ones related to several different traits, such as bone mineral density or cardiovascular traits.

# 1 Introduction

Genome-wide association studies (GWAS) of imaging modalities pose a challenge due to the high dimensionality of the data—for example, structural brain MRI data scans are typically comprised of over 7 million voxels. The naive approach of testing for associations with all elements in an image is infeasible due to both the computational cost and the loss of statistical power when adjusting the p-value thresholds to account for multiple testing against all voxels.

One solution is to extract predefined image-derived phenotypes (IDPs) [1–3], which aggregate spatial information from imaging data into single variables, e.g., volumes or intensities of particular brain regions of interest (ROIs), reducing the number of phenotypes to test against. While being interpretable, such analyses require the availability of automated tools for IDP extraction for the modality of interest and are limited to traits defined a priori, potentially preventing novel genetically-driven phenotypes from being discovered.

Another approach considers all voxels for genetic association testing. However, instead of performing $N_{variants} \times N_{voxels}$ independent univariate tests, all genetic variants and all phenotypes are modelled jointly in order to increase statistical power and decrease the number of computations. vGWAS proposed by [4] is a voxel-wise method which retains only the genetic variant with the lowest p-value per-voxel and estimates the effective number of tests using linkage disequilibrium (LD) information to relax the significance thresholds. The Multivariate Omnibus Statistical Test (MOSTest) [5, 6] additionally accounts for polygenicity, and is able to detect variants with smaller effect sizes. While improving statistical power, both of these approaches require performing computations for each variant-voxel pair. [7] apply a projection into a lower-dimensional space on both the variants and voxels, reducing the computational burden, and allowing to employ permutation testing instead of parametric testing as a means of controlling the familywise error rate.

A recent line of work employed deep learning (DL) to extract imaging features using pretrained DNN models to perform GWAS on, which has been demonstrated to be successful in a range of imaging modalities, including retinal fundus images [8, 9], cardiovascular magnetic resonance (CMR) images [10, 11], or brain MRI scans [12]. This can be seen as a compromise between the IDP and voxel-wise approaches, reducing the dimensionality of the imaging data, while not being limited to a priori defined IDPs.

Here, we employed the transferGWAS method of [8], which consists of 1) training DNN feature extractors on auxilliary datasets 2) extracting features from imaging data in the "target" GWAS dataest using the trained DNN models 3) further reducing the feature dimensionality using principal component analysis (PCA) 4) conducting GWAS on the resulting feature principal components (PCs). We used transferGWAS to perform an imaging GWAS on $N = 36,311$ T1-weighted brain MRI scans from UKB. We encoded the brain scans from UKB using

DNN models trained on the ImageNet [13], and the Alzheimer's Disease Neuroimaging Initiative (ADNI) datasets [14], with the former extracting "general" image features, and the latter focusing on brain MRI and dementia-specific ones. We note that a similar study has been performed on the UKB brain MRI data by [12] using the ENDO approach [9], which reserves a subset of the target dataset for training of the DNN model, therefore reducing the sample size available for GWAS. By employing the transferGWAS method we were able to utilize the full sample size for genetic testing, due to the DNN models being trained on auxilliary datasets.

Our GWAS of 20 DNN features identified 289 associated genetic loci. 21% of them were not previously reported in brain studies, and corresponded to traits such as bone mineral density (BMD), body size, or blood cell counts, indicating connections between these phenotypes and brain structure, while 11 had no previously reported phenotypic associations. A number of the discovered regions was not detected in either the IDP [2] or ENDO [12] brain MRI studies. We further conducted downstream analyses using the discovered genetic variants, demonstrating their utility in creating more predictive PGS, and pointing to novel genetic correlations between type 2 diabetes (T2D) and diffusion magnetic resonance imaging (dMRI) traits (see Fig 1 for an overview of our workflow).

## 2 Results

### 2.1 Extraction of deep features from the T1-weighted UK Biobank brain MRI scans

To obtain the low-dimensional feature representations of the brain MRI scans, we encoded each of the 36, 311 MRI samples using two pretrained DNN models (we will refer to them throughout the text as the ImageNet and the ADNI model). To reduce their dimensionality, we applied PCA on the extracted features, retaining the first 10 PCs for each model, resulting in 20 PCs in total. The choice of 10 PCs was motivated by [8], who found that a larger number of PCs did not lead to substantial improvements in term of discovered variants, and by the fact that the 10 PCs explained 81% and 92% of variance in the ADNI and ImageNet models respectively.

### 2.2 Interpretation of the DNN features

In order to interpret the signal carried by the DNN PCs, we performed a phenome-wide association study (PheWAS) against each PC and 7, 744 UKB phenotypes (S2 Table). We found 2, 408 and 2, 622 significantly associated phenotypes for the ImageNet and ADNI PCs respectively, having p-values below the Bonferroni-corrected threshold of $\approx 6.5 \cdot 10^{-7}$. Fig 2 shows the percentage of significantly associated traits per category. The top 35 categories with the highest ratio of significant hits contained 17 brain-related categories, with the other ones being bone density, body composition, or blood-related categories. In almost all cases the ADNI PCs were associated with a higher number of distinct phenotypes than the ImageNet PCs.

Furthermore, we analyzed how the features correlate with different brain ROIs, using brain segmentation masks obtained with the Synthseg software [15]. For each brain ROI, we computed the fraction of voxels correlated with a given PC (Fig 3), the ratio of explained variance per ROI (Fig I in S1 Text), and per-voxel heatmaps (Figs E, F, G, H in S1 Text). Most PCs were correlated with voxels of most ROI, although at different ratios. The ImageNet PC were correlated on average with a higher number of voxels than the ADNI ones. In particular, the first two ImageNet PCs were correlated with 99% of all ROI voxels, and explained 30% and 7% of the total variance. These two PCs, as well as several other ImageNet PCs seemed to be encoding the overall scan brightness (Fig F in S1 Text). While the ADNI PCs were explaining a

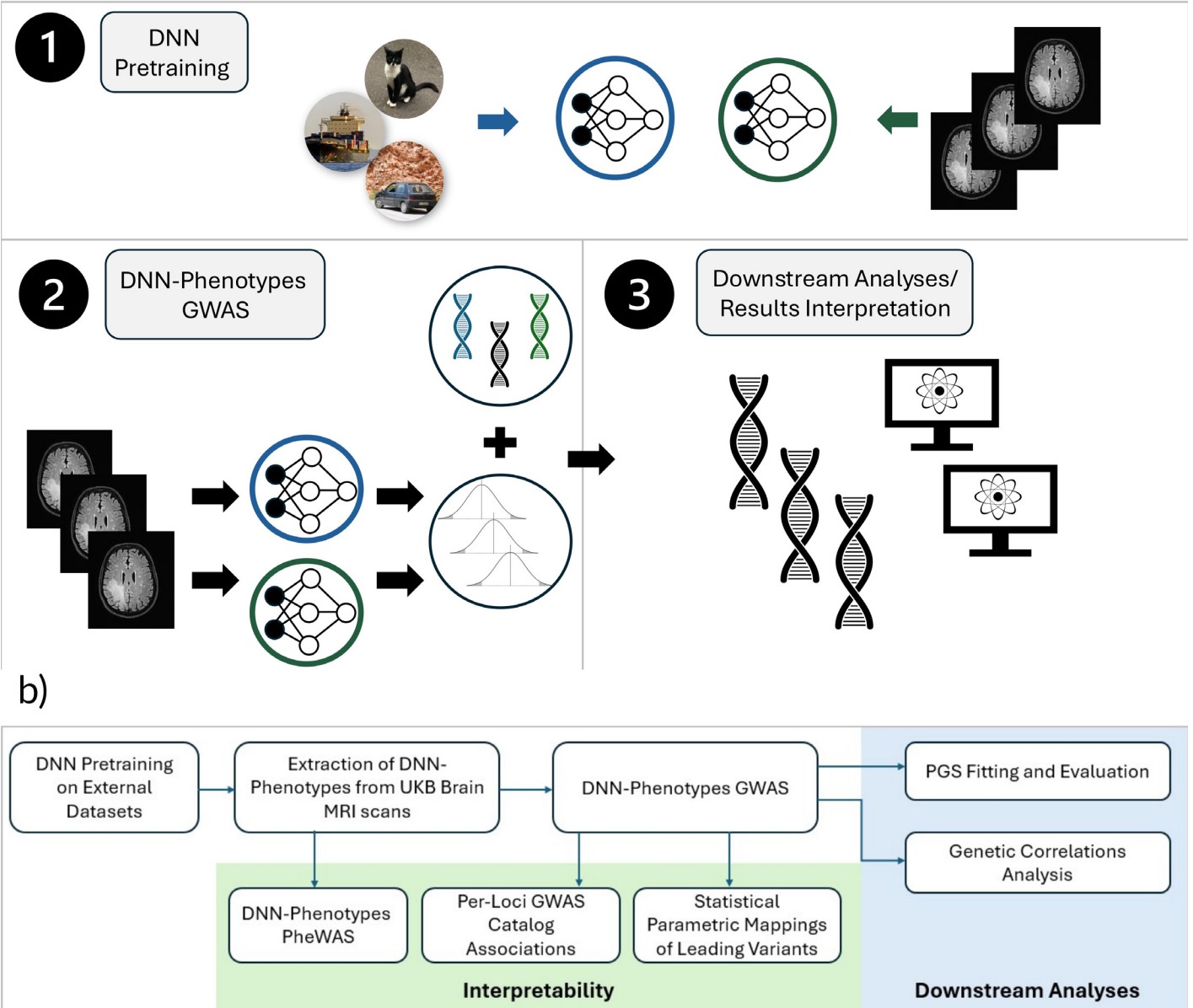

**Fig 1. Overview of our study and workflow. (a)** A general overview of the study: (1)—we trained 2 DNN models on external datasets of natural images, and of brain MRI scans (2)—encoded brain MRI data from the target datasets and performed GWAS on the DNN-derived phenotypes (3) performed a series of downstream analyses using the learned DNN features and discovered genetic variants. **(b)** Description of each step involved in the complete workflow of our study.

smaller amount the total variance in voxel intensity (8% of total variance compared to 42% of the ImageNet model), they seemed to be focused on particular brain structures—e.g., PC 5 corresponding to the lateral ventricles—rather than the overall scan intensity (Figs G, E in S1 Text).

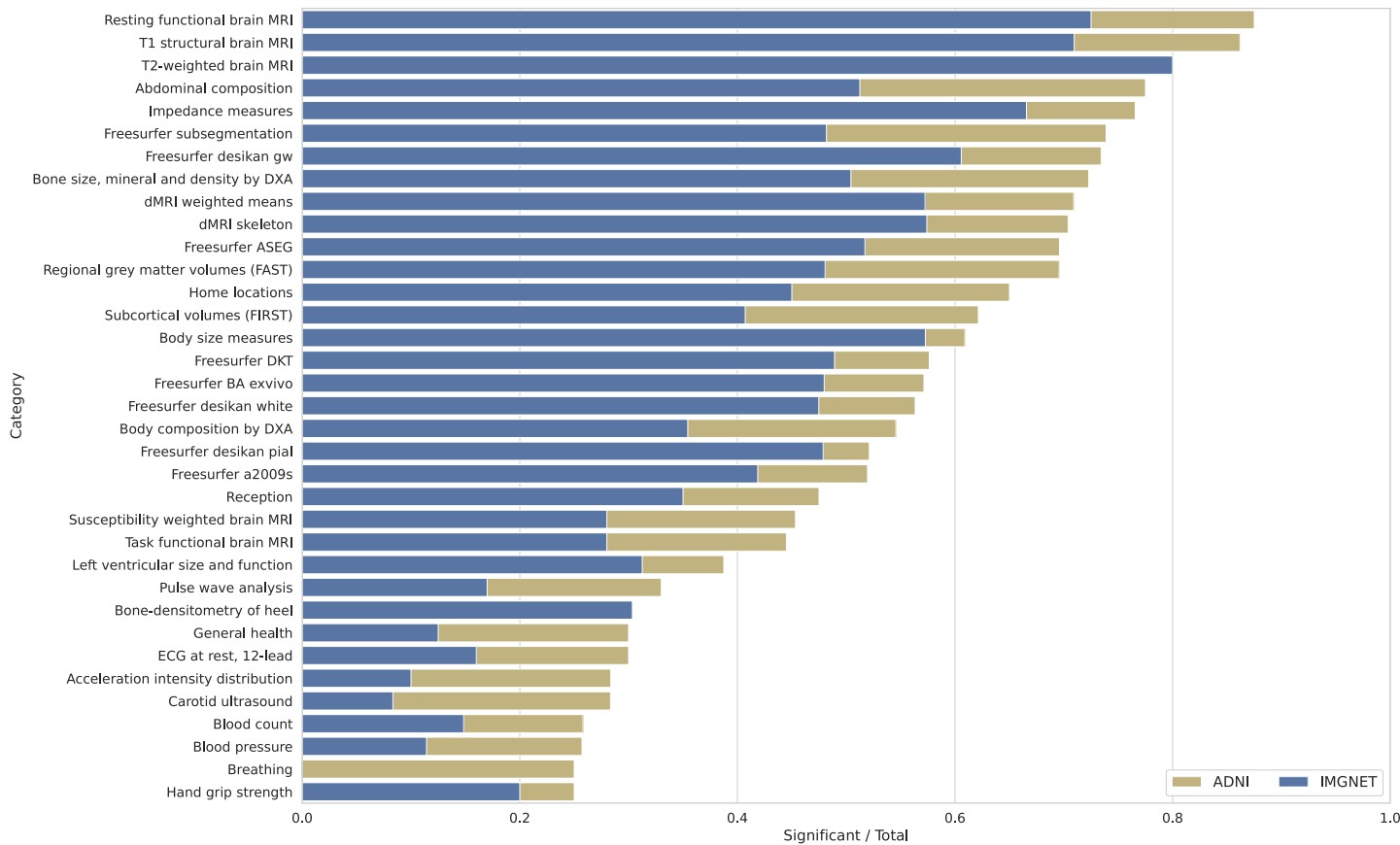

**Fig 2. Results of the PheWAS performed on the principal components (PCs) of the ImageNet (blue) and ADNI (yellow) pretrained models.** For each phenotype category from UK Biobank (UKB) we plot the number of significant associations per model divided by the total number of traits in that category—in case of multiple PCs being associated with a phenotype, we only count them once. Shown are the top 35 phenotype categories with the highest ratio of significant associations.

## 2.3 GWAS results

At the Bonferroni-corrected significance threshold of $2.5 \cdot 10^{-9}$, we found 4, 665 peak associations for the ImageNet and 5, 291 for the ADNI pretrained models, resulting in 4, 382 and 4, 360 distinct variants for ImageNet and ADNI. The clumping procedure then identified 194 and 165 independent regions for the ImageNet and the ADNI models respectively, with 70 regions being shared between the two models (see Fig A in S1 Text for a per- chromosome comparison). This amounted to 7, 458 distinct variants and 289 distinct regions across all 20 features of both DNN models. Figs 4 and 5 show the Manhattan plots for both models, aggregated over each of the 10 PCs per model. We estimated the heritability of each PC using linkage disequilibrium score regression (LDSC) (Section 4.5), and found all PCs to be significantly heritable (p-values below $10^{-8}$), with the ADNI-pretrained PCs having a mean $h^2 = 0.19$, and the ImageNet PCs having a mean $h^2 = 0.13$ (Fig 6). The summary statistics for all PCs are made publicly available as a figshare resource under https://doi.org/10.6084/m9.figshare. 25933717.v1.

**2.3.1 GWAS catalog associations.** For each independent locus, we queried associations reported in previous GWA studies from the NHGRI-EBI GWAS Catalog [16] (Fig 7). The dominating phenotype categories included BMD-related traits and a range of brain traits, such as cortical thickness, diffusion, or volumes of brain ROI. We note that the ADNI-pretrained

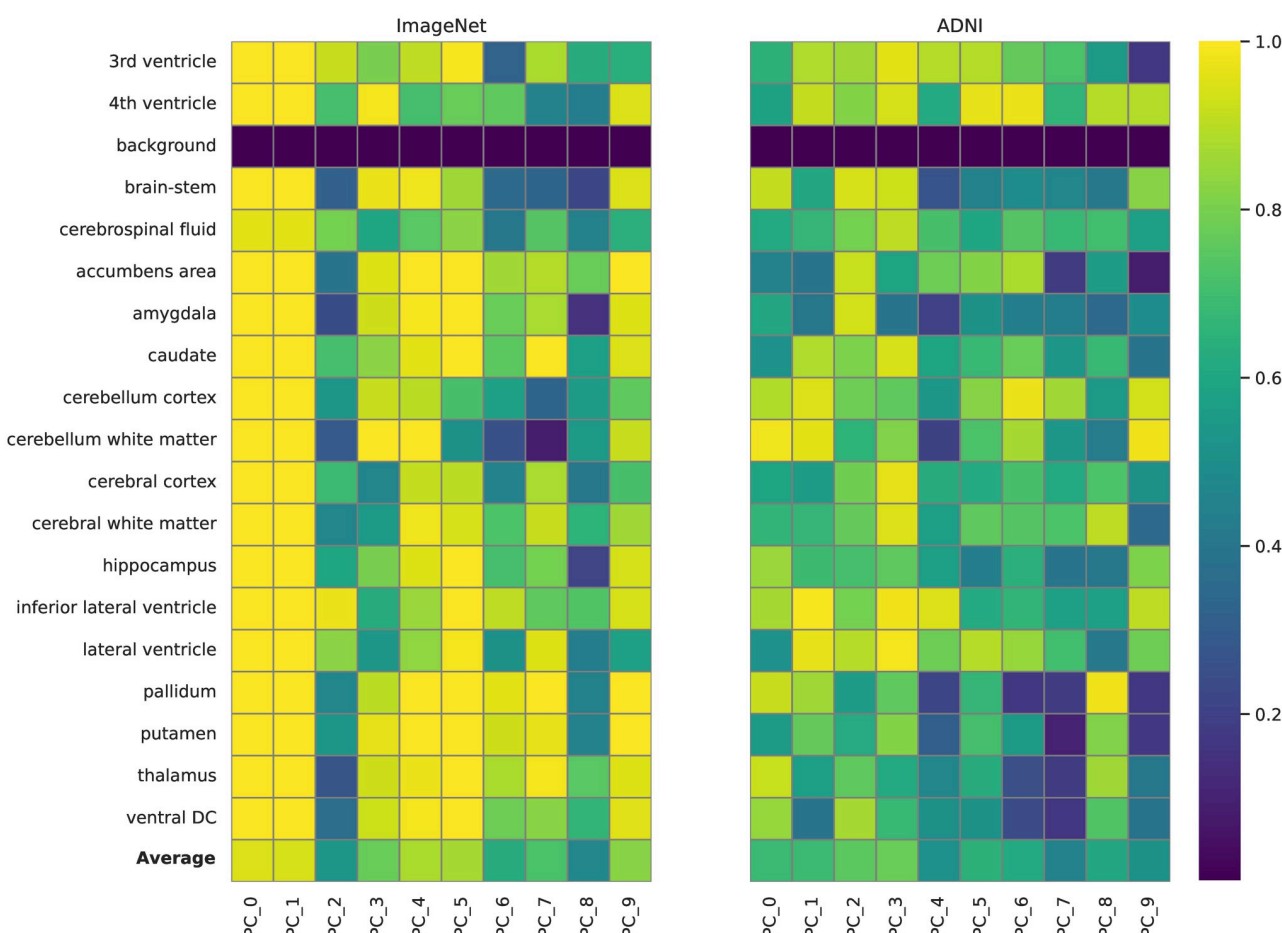

**Fig 3. Fraction of significantly correlated voxels in each brain region of interest (ROI) for each principal component of the neural network models features.** Values are computed as the total number of significantly associated voxels in each ROI divided by the total number of voxels in that ROI.

features tagged more regions corresponding to brain-related traits, whereas the ImageNet model tagged more regions related to "general" body structure, such as BMD, height, or body mass index (BMI). Overall, out of the 289 independent loci, 62 did not have brain-related associations reported in the catalog.

Among neuropsychiatric disorders with the highest number of distinct regions, 47 were associated with schizophrenia, 37 with neuroticism, 36 with attention deficit hyperactivity disorder, 35 with bipolar disorder, 33 with depression, 32 with Alzheimer's disease, 30 with autism, 22 with anorexia nervosa and 21 with anxiety.

3 out of the 10 first traits were not directly brain-related: heel bone mineral density (HBMD) (144 regions), total BMD (125 regions), and height (113 regions). The associations between BMD and the brain have been investigated in the context of neurological disorders [17–19], as well as in samples of healthy subjects [20]. [17, 18] reported a correlation between BMD and an early onset of Alzheimer's disease (AD), as well as with several brain volumes. HBMD is postulated to be a causal factor for multiple sclerosis (MS) through an increased risk of fractures [19]. [20] showed that osteoporosis increases the pace of parenchymal atrophy and ventricular enlargement during aging of healthy individuals.

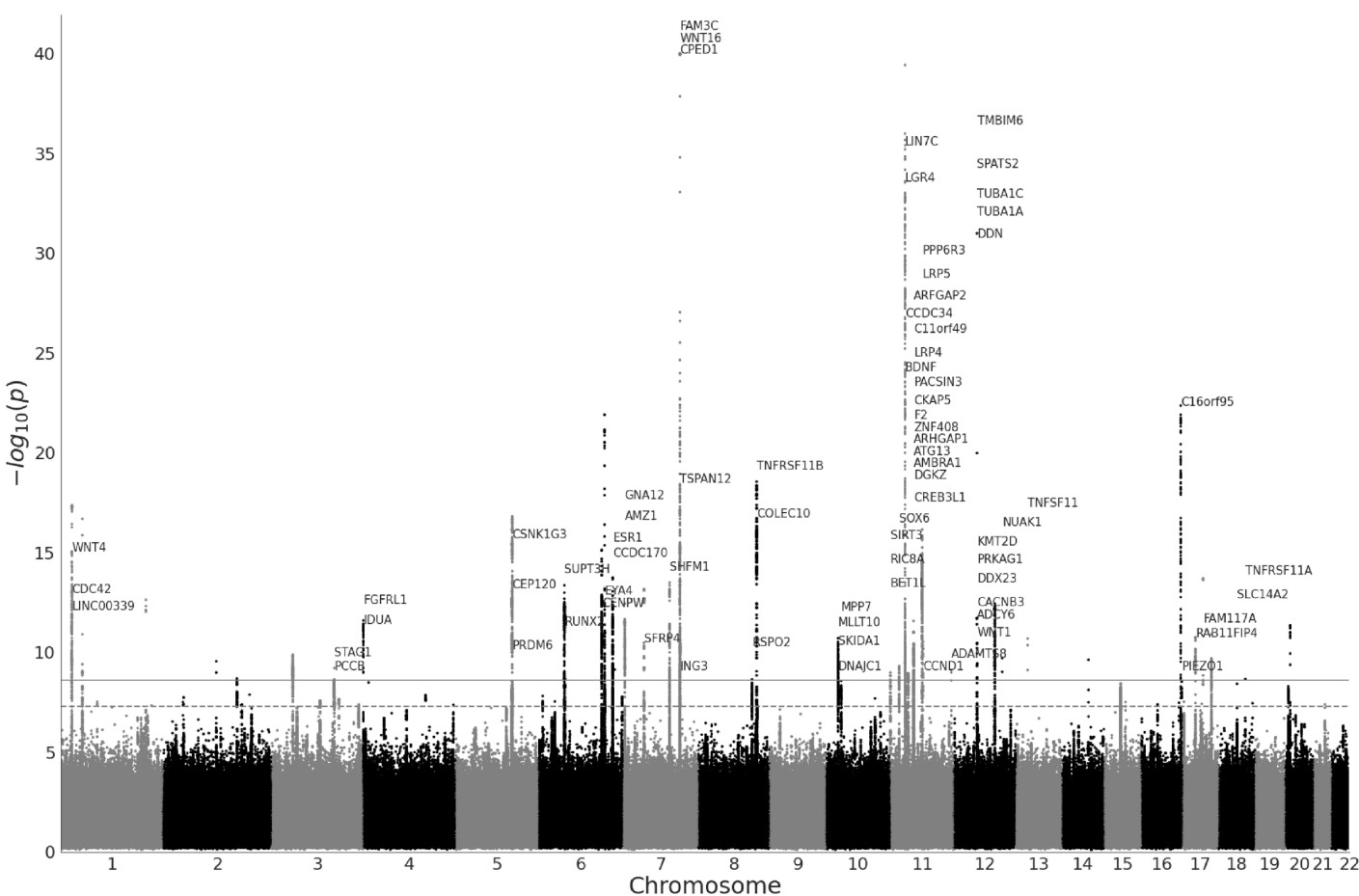

**Fig 4. Manhattan plot of GWAS ($n$ = 36, 311 individuals, 16, 472, 121 SNPs) performed on features of the ImageNet-pretrained model aggregated over all 10 features of the model.** The horizontal lines mark the initial significance threshold of $5 \cdot 10^{-8}$ (dashed line) and Bonferroni-corrected threshold of $2.5 \cdot 10^{-9}$ (solid line). We plot gene names for leading variants in each locus. For visualization purposes we truncate p-values below $10^{-40}$ and plot only the minimal p-values across the 10 features.

Another prevalent category were blood-related traits, such as cell counts: white (67), red (32), monocyte (45), neutrophil (41), eosinophil (40), lymphocyte, (25) reticulocyte (23), blood pressure (95) or hypertension (45), or hemoglobin (68). Blood pressure and hypertension are known factors influencing brain morphology, as well as cognitive performance or dementia [21–24], while aenemia is a causal factor for cognitive decline and AD [25, 26].

**2.3.2 Loci without previously reported GWAS catalog associations.** In total, we found existing phenotypic associations in the GWAS Catalog for 275 regions in studies conducted on the white British population, and 278 regions among all populations, whereas 11 loci had no previously reported associations. 8 of these loci were located inside 6 distinct gene regions: CPED1, WNT16, TSPAN12, RP11–161H23.9, WNT1, and C16orf95. CPED1, WNT16, and TSPAN12 have recently been identified as a region of BMD genes [27]. WNT1 is protein-coding gene with variants associated with BMD or cognitive function [28, 29]. C16orf95 is a protein-coding gene with associations with a range of brain measurements and BMD [30, 31]. RP11–161H23.9 is a differential expression long non-coding RNA gene whose expression was associated with glioblastoma survival [32]. The remaining 3 loci not residing within genes were located in regions identified as enhancers in the Enhancer Atlas [33]. In particular, the

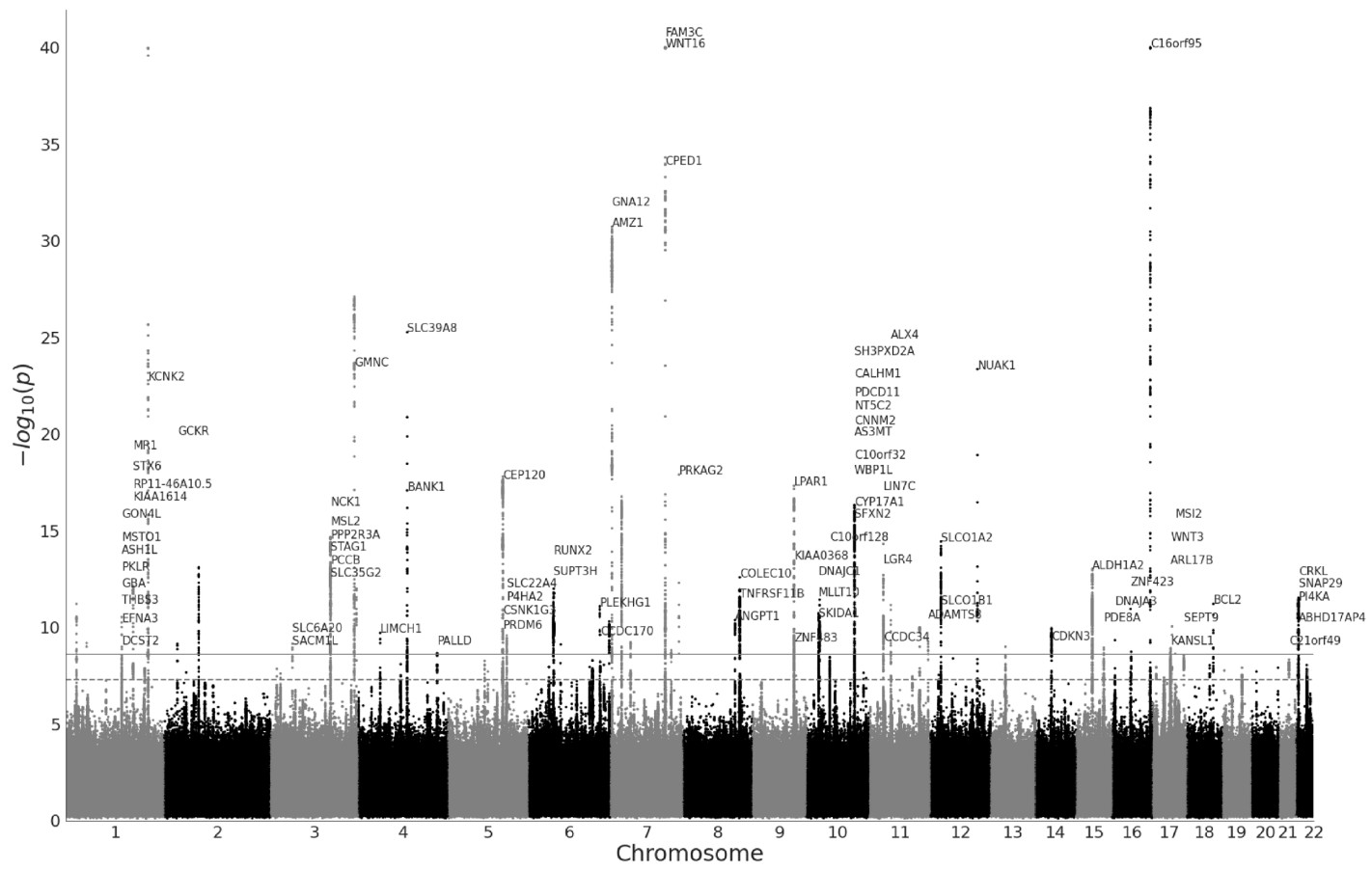

**Fig 5. Manhattan plot of GWAS ($n$ = 36, 311 individuals, 16, 472, 121 SNPs) performed on features of the ADNI-pretrained model aggregated over all 10 features of the model.** The horizontal lines mark the initial significance threshold of $5 \cdot 10^{-8}$ (dashed line) and Bonferroni-corrected threshold of $2.5 \cdot 10^{-9}$ (solid line). We plot gene names for leading variants in each locus. For visualization purposes we truncate p-values below $10^{-40}$ and plot only the minimal p-values across the 10 features.

loci around 6:156060806 and 7:121093894 had markers for enhancer activity in human dendritic, fetal brain, and neural stem cells.

As a further means of interpreting the novel regions, we computed per-voxels correlations for each leading single-nucleotide polymorphism (SNP) (Fig 8), and the fractions of volume of each brain region correlated with each lead variant (Fig 9). All SNPs were correlated with the left cerebral cortex and cerebrospinal fluid. Most notable was the variant rs111469125 (16:87268090) located inside the C16orf95 gene, being correlated with 19 out of 22 brain regions, in particular with several ventricle structures: the 3rd and 4th ventricles (3% of total voxels), and the left and right lateral ventricles (1% and 1.5% of total voxels). It was also correlated with 3% of the voxels of cerebrospinal fluid, and was the only new variant correlated with the left cerebellum white matter, the left thalamus, and the right caudate. On the other hand, rs546521618 (6:156060806) might be a potential false positive, being the only variant in its loci, and having a minor allele frequency (MAF) below 1.3%

**2.3.4 Comparison with previous studies.** We performed another GWAS on the UKB sample, splitting it into discovery and replication cohorts (23,604 and 12,709 samples), replicating 1,631 hits over 1,510 unique variants, which amounted to 70 replicated loci. We compared our results with two GWA studies on UKB brain MRI data—the first one using 3,144

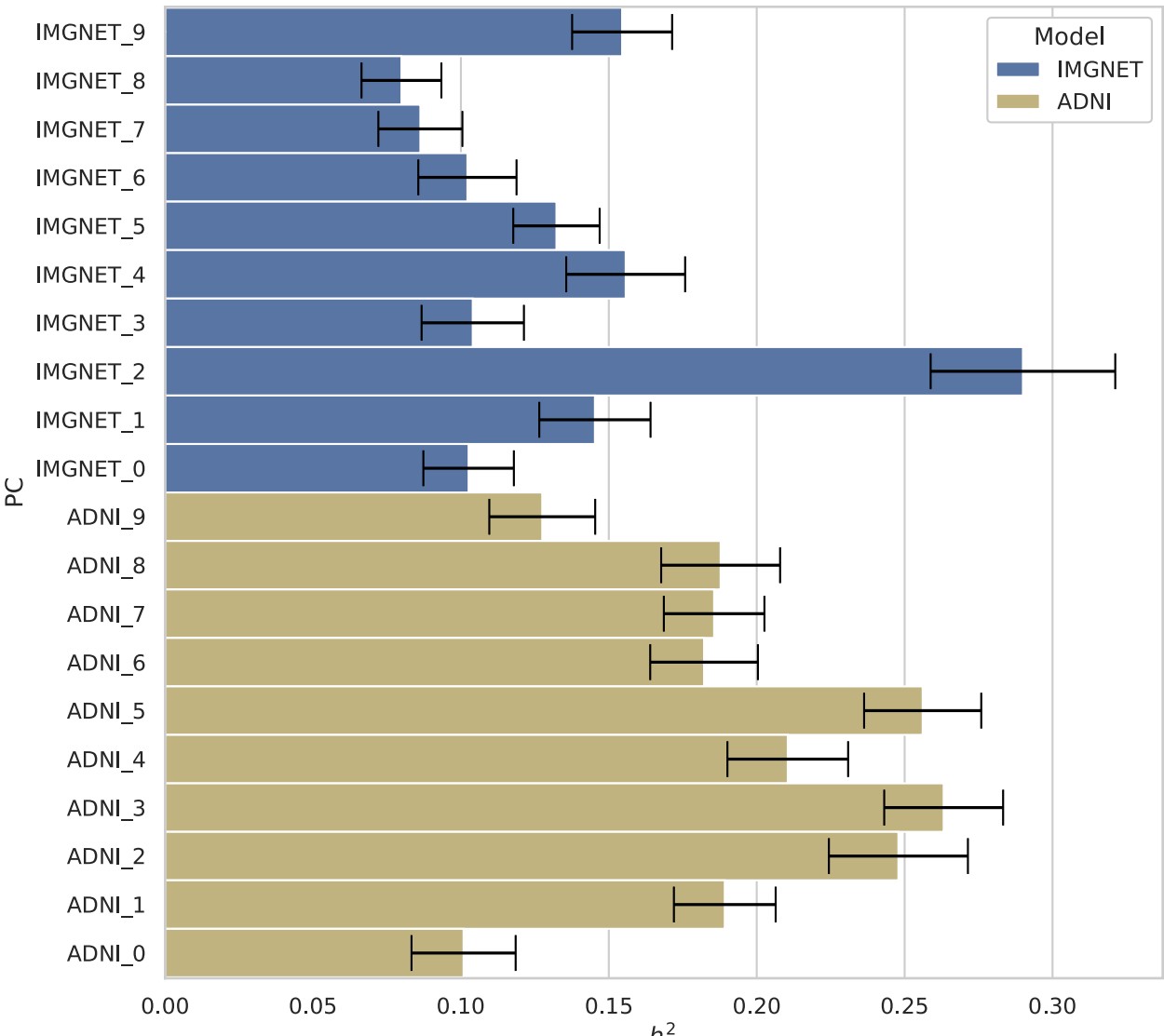

**Fig 6. $h^2$ heritability estimates of principal components (PCs) of the ImageNet (blue) and ADNI (yellow) pretrained models, obtained using linkage disequilibrium score regression (LDSC).** Black lines indicate the standard error of the estimates.

brain imaging-derived phenotypes [2] and the second study using 256 DL-based features [12], which yielded 692 and 43 replicated loci respectively. Out of our 70 replicated loci, 9 were not present in the 692 of [2], and 28 were not present in the 43 loci of [12].

## 2.4 TransferGWAS polygenic scores

Here, we evaluated the potential of variants discovered in our study for downstream prediction of phenotypes using 20 PGS fitted for each of the 20 DNN PCs with the summary statistics from our GWAS. In order to compute the features of the DNN models, imaging data need to be present, which constitutes less than a tenth of all UKB samples. On the other hand, genotyping data were available for all participants. This allowed us to calculate the PGS for all remaining $N = 424, 609$ white British participants not included in the GWAS sample. The

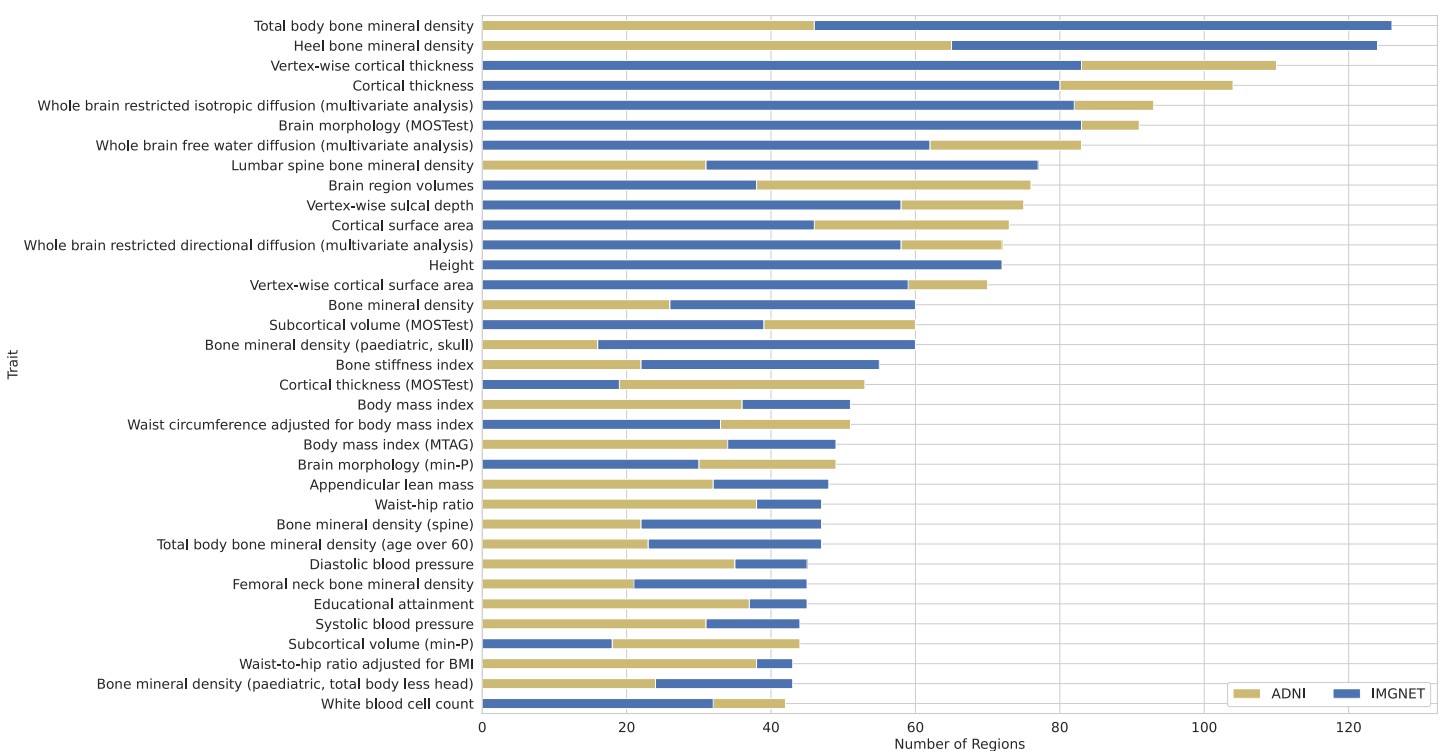

**Fig 7. Number of independent loci per trait with associations reported in previous studies included in the NHGRI-EBI GWAS Catalog [16].** Shown are the top 35 traits with the highest number of associated regions.

corresponding methods are described in Section 4.4, while the weights of the fitted scores are made publicly available as a figshare resource under https://doi.org/10.6084/m9.figshare.25933663.v1.

**2.4.1 PGS PheWAS.** To gain insights into which traits the PGS might be predictive of, we performed a PheWAS on the 20 PGS and the 7,744 UKB phenotypes (S1 Table). Note that while the "raw" DNN PCs can encode both genetic and environmental signals, the PGS should capture only the former, and thus we expected the associations between the phenotypes to differ from the PheWAS performed on the PCs. The total number of significant PC-phenotype associations and the effect sizes were higher for the original PCs than for the PGS: 28,767 vs. 13,199 significant associations in total, 2,928 vs. 1,537 distinct associated traits, with mean effect sizes of $\bar{\beta} = 0.08$ vs. $\bar{\beta} = 0.04$. We identified 3 potentially interesting groups of associations (Fig C in S1 Text):

- traits related to BMD or bone fractures

- height, weight, BMI

- cardiovascular traits, and blood biomarkers

which we decided to investigate further in a prediction setting.

**2.4.2 Predictive performance compared to trait-specific PGS.** We tested the utility of our developed PGS by evaluating whether they can improve predictions of phenotypes from UKB over PGS designed specifically for particular traits in a multi-PGS setting [34]. We chose a set of 9 phenotypes based on the PheWAS results and computed their corresponding scores using PGS available in the PGS Catalog [35]. For each phenotype, we then fitted and evaluated

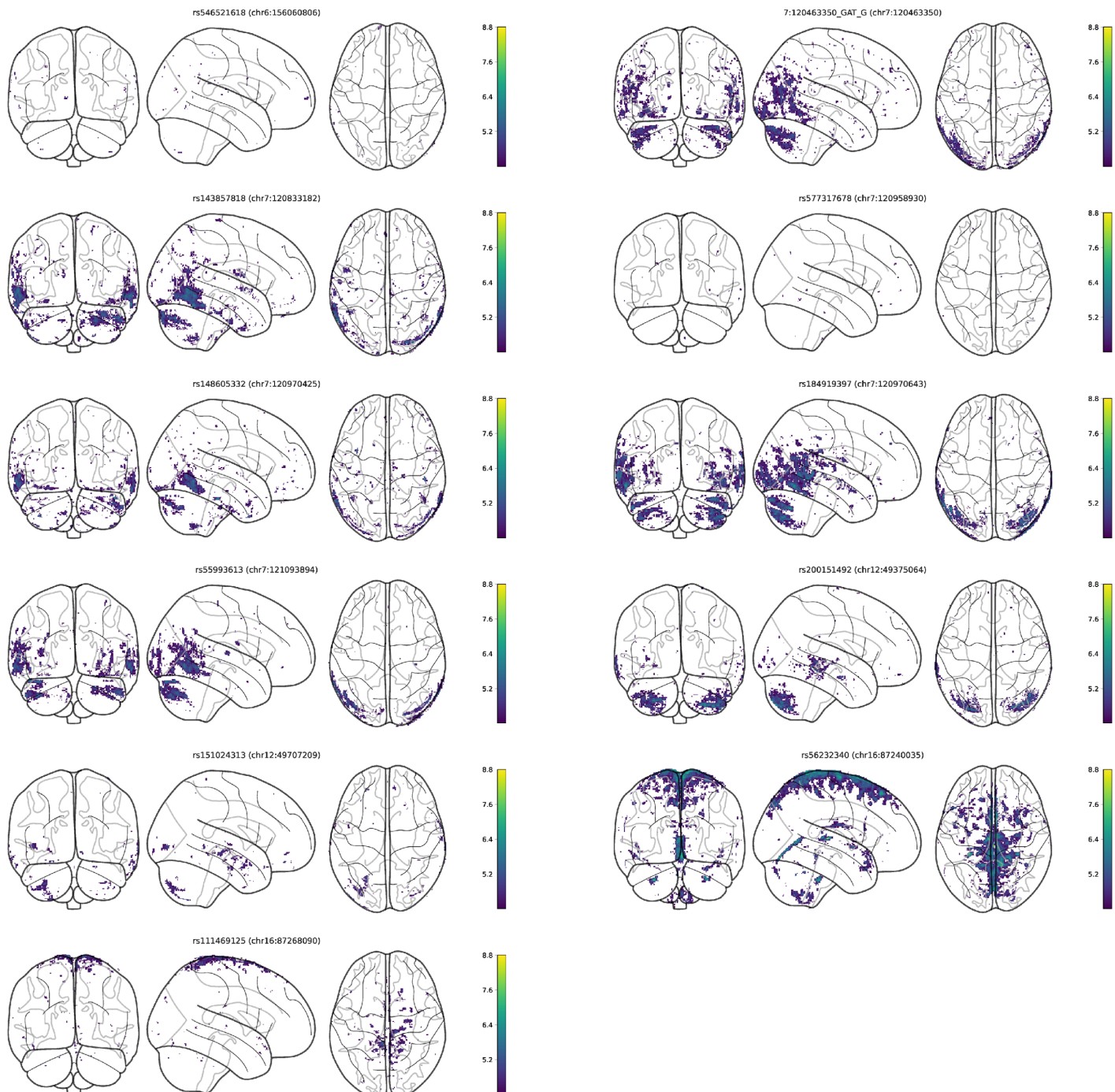

**Fig 8. Brain MRI voxels corresponding to genetic regions with no previously reported GWAS associations.** Plotted are values of the *t*-statistics of the correlation coefficients between lead variants of each region and each single voxel in the MRI scans. We plot values below the Bonferroni-corrected significance threshold accounting for the total number of voxels tested.

two linear models: one fitted using only the trait-specific PGS, and one additionally using our transferGWAS PGS. While there were statistically significant improvements in predictions for 6 out of 9 traits, they yielded arguably small performance increases ($\sim 1.5\%$ of relative improvement), with the exception of predicting HBMD using a (general) BMD PGS, where

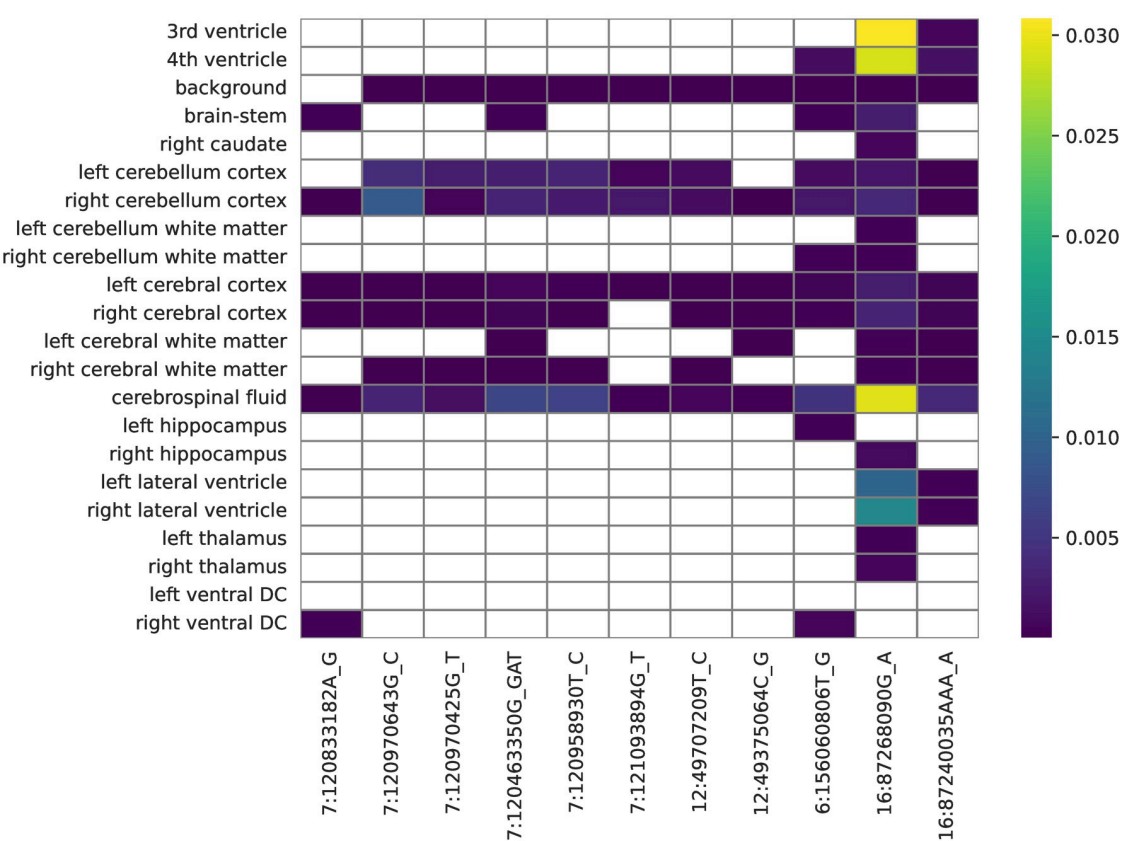

**Fig 9. Fractions of volume of brain regions correlated with genetic regions with no previously reported GWAS associations.** The values are computed as the total number of voxels in a given brain region significantly correlated with a lead variant, divided by the total number of voxels in that brain region. White cells indicate no voxels being significantly correlated for a given brain region-genetic region pair.

the relative improvement was over 24% (Table 1). We decided to further investigate the HBMD results. Since the improvement could have been stemming from a lower signal in the dataset of the external PGS, compared to the UKB, we conducted a further comparison within the UKB. We performed a GWAS on HBMD using 19, 909 samples from our GWAS data

**Table 1. Comparison of predictive performance of Multi-PGS models on the white British population of UK Biobank (UKB) using only trait-specific polygenic scores (PGS) (2nd column) and including our TransferGWAS PGS (3rd column) for a set of selected phenotypes from UKB, measured with the $R^2$ coefficient of determination.** Significant differences are marked with (*). Heel bone mineral density (1) and (2) correspond to results of using PGS for heel bone mineral density, or (general) bone mineral density respectively. Statistical significance was estimated using pairwise permutation tests with 1, 000 permutations.

| Trait | PGS Catalog | Our PGS + PGS Catalog | Δ |
|---|---|---|---|
| Height | 0.760 | 0.760 | 0.00008* |
| BMI | 0.286 | 0.286 | 0.000 |
| Heel bone mineral density (1) | 0.274 | 0.277 | 0.002* |
| Heel bone mineral density (2) | 0.062 | 0.077 | 0.015* |
| Red blood cell count | 0.389 | 0.389 | 0.000 |
| White blood cell count | 0.097 | 0.099 | 0.002* |
| Systolic blood pressure | 0.283 | 0.283 | 0.0002* |
| Diastolic blood pressure | 0.175 | 0.176 | 0.001* |
| Ventricular rate | -0.001 | -0.005 | -0.004 |

which had HBMD measurements available, and 16, 404 randomly drawn from the remaining samples of the white British participants, to match the sample size and population of our GWAS, and fitted a new PGS on the resulting summary statistics. We then evaluated the newly-created HBMD PGS with and without our transferGWAS PGS on the remaining UKB data and observed the same relative improvement of 1% in performance ($p < 0.001$). This indicates that transferGWAS has the potential to identify additional variants for related traits while using the sample size. We hypothesize that this might be due to certain pleiotropic variants having a larger effect on the DNN PCs than on HBMD, and thus being able to be detected with our DL GWAS and not with the HBMD-dedicated GWAS.

## 2.5 Genetic correlations

The results of the PheWAS conducted on the learned PCs led us to a set of traits that we decided to investigate further. In order to analyze the genetic components of the PCs, we computed genetic correlation coefficients between 102 selected traits and each of the 20 PCs (see Section 4.5 for details). 39 traits were significantly genetically correlated, surpassing the Bonferroni-corrected threshold of $\approx 2.5 \cdot 10^{-5}$. We grouped the traits into 3 groups:

- (volumes of) brain ROI (e.g., ventricles, brain stem, cerebrospinal fluid (CSF))

- dMRI traits (e.g., fractional anisotropy (FA), orientation dispersion index (ODI))

- "general" traits: Height, T2D, BMI, HBMD

Additionally, we tested for correlations with AD, educational attainment, and unipolar depression, finding no significant correlations when corrected for multiple testing ($p > 0.001$). The significantly associated traits are shown in Fig 10, where we observed several "clusters" of PC-trait associations.

Several PCs were genetically correlated with volumes of multiple brain ROI. The first two PCs of ImageNet (IMGNET0, IMGNET1) seemed to capture the overall body size, as they were negatively genetically correlated with height and white matter volume, and positively with ventricular ROI and CSF.

PCs ADNI2, ADNI3, and IMGNET4 were genetically associated with volumes of several brain ROI, e.g., cerebral white matter, putamen, or thalamus. ADNI2 and ADNI3 were also genetically associated with volumes of CSF and the lateral ventricle. Interestingly, ADNI2 had a positive genetic correlation both for CSF and the lateral ventricle, as well as for gray and white matter structures, whereas one might expect the ventricular volumes (and thus CSF) to grow with the shrinkage of brain structures.

PCs genetically associated with HBMD seemed to capture different aspects of brain anatomy. IMGNET2 had a negative genetic correlation with HBMD, BMI, and cerebral white matter, but also with multiple ventricular volumes. On the other hand, ADNI8 and IMGNET4 also had negative genetic correlations with HBMD, but positive ones with cerebral white matter.

ADNI0 and ADNI4 were genetically associated with a range of Diffusion MRI traits, as well as with several ventricular ROI. Furthermore, ADNI4 was genetically correlated with HBMD and BMI, and was the only PC genetically associated with T2D, which we further discuss below.

**2.5.1 ADNI4 and T2D.** BMI was shown to increase the risk of developing T2D [36, 37], as well as being genetically correlated to T2D [36]. The signs of genetic correlations between ADNI4, and BMI and T2D were also matching. ADNI4 was also positively genetically associated with HBMD. T2D patients have been shown to have a higher bone density [38, 39].

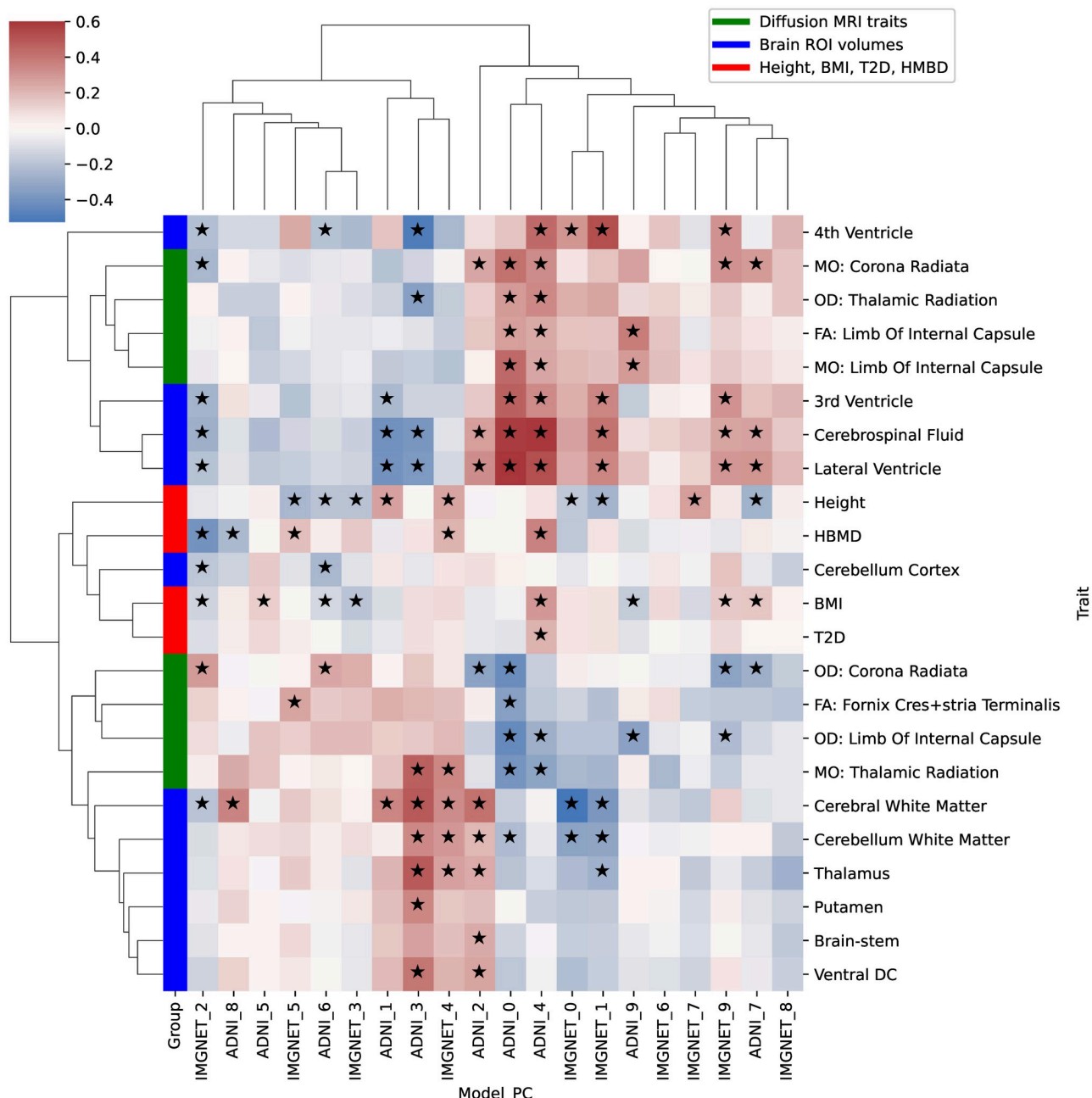

**Fig 10. Genetic correlation coefficients between the 20 deep neural network (DNN) principal components (PCs) (rows) and 23 significantly associated phenotypes (columns), out of 27 candidate traits from the UK Biobank (UKB).** Cell colors represent the magnitudes and the signs of the estimated genetic correlation coefficients between each PC and phenotype combination.

Evidence also exists for shared heritability between BMD and T2D, albeit relatively small [40, 41]. As with BMI, the sign of the genetic correlation between ADNI4 and HBMD was positive. Regarding the brain ROI, ADNI4 was positively genetically correlated with volumes of the lateral, 3rd, and 4th ventricles, as well as with the CSF. Ventricular enlargement and increase in CSF are associated with several neurodegenerative diseases, such as AD, MS, or schizophrenia [42, 43]. Several studies showed an association between T2D and volumes of white matter

structures (whole brain volume, frontal lobe), gray matter (overall trend in all structures), as well as CSF and ventricular volumes [44]. Furthermore, ADNI4 was genetically correlated with 35 different dMRI traits:

**Mean diffusivity (MD) traits**. 4 MD traits were positively genetically correlated with ADNI4: fornix, superior serebellar peduncle (both sides), and the superior fronto-occipital fasciculus (left). Positive associations between T2D and MD have been found in observational studies [45, 46].

**Fractional anisotropy (FA) traits**. FA traits have been found to be negatively correlated with T2D in literature [44–46]. We found 4 traits to be negatively genetically correlated with ADNI4, however the posterior limb of left internal capsule was positively genetically correlated with the PC. The direction of this correlation seemed to be in opposition to the associations found in observational studies [44]. On the other hand, it is postulated to be causal with the same sign for fasting insulin [47], an increase of which is an indicator of T2D. We identified two regions containing shared variants located at Chr2:27766284 and Chr14:91881387. The first region contains missense and intron variants for GCKR gene (ENSG00000133962), a glucokinase regulator, with no previously reported associations for brain phenotypes, missense and intron variants for C2orf16 (ENSG00000221843) and intron variants for ZNF512 (ENSG00000243943) both protein coding genes with association with neurodegenerative diseases, T2D, and blood measurements. The second region contains intron variants for the CCDC88C (ENSG00000015133), a protein coding gene, with associations with glucose metabolism, brain measurements, and neurodegenerative diseases, and CCDC88C-DT (ENSG00000258798), an RNA gene that is a divergent transcript for CCDC88C, with associations with brain measurements and hypertension. The above may be another indicator of a non-trivial relation between FA of limb of internal capsule and T2D, with potentially different shared heritability and environmental effects.

**Orientation dispersion index (ODI) traits**. 7 ODI traits were positively genetically correlated with ADNI4, while 3 traits were correlated negatively. ODI of white matter tracts was reported to be positively correlated with duration of T2D and with levels of HbA1c, a marker for T2D, while ODI of internal capsule was reported to have a negative correlation [48], which is consistent with 9 out of 10 of our findings. We found a negative genetic correlation for the posterior right corona radiata, which had shared variants in regions located at chr8:119486034 and 11:27465591. The first region has intron variants for SAMD12 (ENSG00000177570), a protein coding gene with associations with brain measurements, MS, bone density, and blood measurements. The second region has intron variants for LGR4 (ENSG00000205213), a protein coding gene with associations with brain measurements, bone density, and body mass traits.

**Mode of anisotropy (MO) traits**. 5 MO traits were genetically positively correlated with ADNI4, and 6 negatively. Fasting insulin, a marker for T2D was reported to be negatively associated with anterior corona radiata [47]. We found positive genetic correlations between both sides of the posterior and superior corona radiata and ADNI4, with shared variants with T2D located in the region chr2:27766284 for the superior, and in chr8:119486034 for the posterior. The first region contains missense and intron variants for the GCKR and C2orf16 genes, and an intron variant for ZNF512 (see the FA regions), and the second region has intron variants for SAMD12 (see the ODI regions above). The correlations between the other 7 traits are reported in S3 Table.

## 3 Discussion

Using the transferGWAS approach we performed a GWAS on 20 DNN feature representations of 36, 311 T1-weighted brain MRI scans from the UKB, identifying 289 loci, 11 of them

without any previously reported associations, and 72 without any associations for brain-related traits. Similar to the findings of the initial transferGWAS study of retinal fundus images of [8], the features of an ImageNet-trained model were associated with a higher number of loci related to "general" body structure traits, such as BMD or BMI, whereas features from a model trained directly on brain MRI data identified more loci corresponding to brain measurements and neurodegenerative diseases. Overall, features of both DNN models were associated directly, through PheWAS, or genetically, through GWAS-identified loci, with a large number of BMD traits. For example, the ImageNet and ADNI-derived features were significantly associated with over 50% and 70% of phenotypes under the UKB category 125 "Bone size, mineral, and density by DXA", and with over 120 and 40 distinct loci associated "Total body bone mineral density" in the NHGRI-EBI GWAS Catalog. Detecting these genetic regions in features derived from brain MRI data seems to confirm the connections between BMD and brain measurements, as well as with neurodegenerative diseases previously reported in the literature (as discussed in Section 2.3.1), which we further investigated with an analysis of genetic correlations (Section 2.5), highlighting particular brain ROI genetically associated with BMD. Furthermore, the genetic correlations identified by our study shed more light on the relations between dMRI measurements and T2D, BMI, as well as cardiovascular traits, also reported in several studies (Section 2.5.1). Finally, we demonstrated a practical application of our findings by constructing PGS of our DNN-derived phenotypes, which improved predictions of existing PGS of BMD, white cell blood count, or diastolic blood pressure. In a further analysis, we fitted a PGS directly to HBMD measurements on a UKB sample of the same size as our GWAS and observed the same improvement in performance when augmented with our DNN PGS, indicating that the transferGWAS approach can identify additional variants for a trait of interest, being complementary to conducting a trait-dedicated GWAS.

We demonstrated how transferGWAS can be applied to discover new variants and in turn, lead to better phenotype predictions. However, a drawback of using features of pretrained DNN models as traits of interest is their reduced interpretability compared to predefined phenotypes. While we analyzed both the DNN-derived traits and the discovered loci with a range of techniques (PheWAS, querying the GWAS Catalog, statistical parametric mappings (SPMs)), we highlight the need for further developing apossibly automated pipeline for interpretability of the DNN features, to foster their utility for consecutive research and clinical applications.

## 4 Materials and methods

### 4.1 Training of the neural network models

The first model used for feature extraction was trained on 4,480 T1-weighted scans from the ADNI dataset [49]. The network architecture was a 3D convolutional variational autoencoder (VAE) [50], trained in a multi-task manner. The model consisted of 3 sub-networks: an encoder, a decoder, and a prediction head. The 128-dimensional outputs of the encoder network constituted the latent representations of the input data. The first task was the standard VAE objective, i.e., reconstructing the input scans from the latent representations, while regularizing the representations to match a standard normal prior distribution with a Kullback-Leibler divergence (KLD) loss term. The second task was to predict the clinical dementia rating (CDR) from the latent representations. The aim of the VAE objective was to learn general structural features describing an MRI scan, while the prediction task should promote neurodegenerative features associated with the presence of dementia. Additionally, we input the age and sex of each participant into the decoder and prediction networks, forcing the model to learn latent representations invariant to age and sex, and thus potentially increasing the

statistical power of the GWAS. We trained the model for 500 epochs with the Adam optimizer [51], with a mini-batch size of 128. The weights of the reconstruction, KLD, and the predictions loss terms were 1, $10^{-4}$ and $10^{-2}$ respectively. For data preprocessing, we skull-stripped each scan using using the HD-BET tool [52], performed a non-linear registration to the MNI152 template with a 1mm$^3$ resolution using the FLIRT and FNIRT commands from the FSL software [53], and finally downsampled the scans to a size of 96×96×96 voxels each.

Following [8], we also employed a 2D ResNet50 [54] model trained on ImageNet, a non-medical dataset of natural images [13]. We used a readily available trained model from the PyTorch library [55]. We selected the 2048-dimensional output of the penultimate layer as the latent features used for the GWAS. Since the model was trained on 2D data, we could not directly extract features from the 3D MRI scans. Instead, for each scan, we computed the features over each single slice across the axial axis and averaged the results into a single vector.

### 4.2 GWAS

We selected a sample of $N$ = 36, 311 UKB participants who "self-identified as white British and have very similar genetic ancestry based on a principal components analysis of the genotypes" (UKB field 22006). We performed the association testing within the linear mixed model (LMM) framework using the BOLT-LMM software [56]. We adjusted for confounding using age, sex, the identifiers of the genotyping array and UKB assessment center, and the first 10 genetic principal components. We filtered the SNPs with the following criteria: MAF≥0.1%, Hardy-Weinberg Equilibrium with a significance level of 0.001, pairwise LD-pruning with $R^2$ = 0.8, and maximum missingness of 10% per SNP and participant, which resulted in 577, 570 directly genotyped SNPs. Including imputed genotype data resulted in 16,472,121 variants in total, on which we performed the GWAS. We clumped the variants into independent loci using the PLINK software [57], with a physical distance threshold of 250kb and a significance threshold of $10^{-9}$ for the index SNPs. We queried the NHGRI-EBI GWAS Catalog [16] using the LDtrait web application [58], with an $R^2$ cutoff of $10^{-1}$ and a 250kb window.

### 4.3 PheWAS

We performed the PheWAS on the PCs of both DNN models using the PHESANT software [59], with a P-value threshold of $\approx 6.5 \cdot 10^{-7}$ from the Bonferroni correction to account for 20 PCs and 7, 744 different phenotypes from UKB, adjusting for age and sex. We note that PheWAS automatically determines the choice of the appropriate regression model (e.g., linear regression for continuous variables or logistic regression for categorical ones) based on the phenotype metadata.

### 4.4 Polygenic scores

We fitted the DNN PGS and the custom HBMD PGS using the PRScs method [60], with the prspipe software [61, 62]. For the predictive performance comparison, we queried the PGS Catalog [35] API for a list of PGS developed for each of the 9 phenotypes, ignoring scores that used the UKB for development, to avoid data leakage. We then computed scores for the $N$ = 451, 450 participants who were not in our GWAS sample using the PGS Catalog Calculator [63]. For each phenotype, we fitted a baseline linear model using all corresponding trait-specific PGS and covariates (age, sex, UKB assessment center, UKB genotyping batch, all UKB genetic PCs) and another linear model which additionally included our 20 DNN PGS. We used 60% of the data for model fitting and evaluated it on the remaining 40%. We computed P-values for differences between achieved $R^2$ scores of the two linear models using

permutation tests with 1, 000 permutations, randomly selecting predictions from either model for each test sample in each permutation.

### 4.5 Genetic correlations

To compute the genetic correlation scores between the PCs and selected traits, we used the LDSC method [64, 65]. We used the provided LD scores precomputed on 1000 Genomes data [66] over HapMap3 [67] SNPs, and used the default values for other parameters of the LDSC. In order to find regions potentially contributing to the genetic correlations between ADNI4, T2D, and dMRI traits (Section 2.5.1), we selected SNPs with a P-value below 0.0001 for which the magnitude of the product of the z-scores between both ADNI4 and T2D, and ADNI4 and a dMRI trait exceeded a threshold of 15. For the dMRI traits, we selected pairs where the sign of the product of the z-scores matched the sign of the genetic correlation with ADNI4. We consider a region a set of variants within 250, 000 base pairs from a "central" variant.

## Supporting information

**S1 Text. Supporting Material main text.** Fig A: Distribution of the p-values from the GWAS from ADNI and ImageNet model training. Fig B: Venn diagrams of the number of discovered loci on each chromosome. Fig C: Results of the phenome-wide association study between the polygenic scores fitted on the features of the deep learning models (rows) and phenotypes from the UK Biobank dataset. Fig D: Results of the PheWAS between the polygenic scores fitted on the features of the DL models (rows) and phenotypes from the UK Biobank dataset. Fig E: Difference images for the brain MRI scans sorted according to their corresponding values of the principal components (PCs) of the ADNI trained model. Fig F: Difference images for the brain MRI scans sorted according to their corresponding values of the PCs of the ImageNet trained model. Fig G: Statistical parametric mappings (SPMs) for the first 10 principal components of the model trained on the ADNI dataset. Fig H: Statistical parametric mappings (SPMs) for the first 10 principal components of the model trained on the ImageNet dataset. Fig I: Ratio of variance explained in each brain region of interest (ROI) by each principal component of the neural network models features. Table A: Comparison of predictive performance of Multi-PGS models on the full UKB sample. Table B: Pearson correlation coefficients of each of the 10 PCs of the ADNI-trained variational autoencoder (VAE) model and age and sex covariates on the UKB data.
(PDF)

**S1 Table. Results of PheWAS on the Polygenic Scores.**
(XZ)

**S2 Table. Results of PheWAS on principal components of the trained model features.**
(TSV)

**S3 Table. Additional results of the genetic correlations analyses.**
(XLSX)

**S4 Table. Description of columns of S1 and S2 Tables.**
(XLSX)

## Acknowledgments

This research has been conducted using the UK Biobank Resource under Application Numbers 77717 and 78537.

## Author Contributions

**Conceptualization:** Alexander Rakowski, Remo Monti.

**Data curation:** Alexander Rakowski, Remo Monti.

**Formal analysis:** Alexander Rakowski, Remo Monti.

**Funding acquisition:** Christoph Lippert.

**Investigation:** Alexander Rakowski, Remo Monti.

**Methodology:** Alexander Rakowski, Remo Monti.

**Software:** Alexander Rakowski, Remo Monti.

**Supervision:** Christoph Lippert.

**Validation:** Alexander Rakowski, Remo Monti.

**Visualization:** Alexander Rakowski.

**Writing – original draft:** Alexander Rakowski, Remo Monti.

**Writing – review & editing:** Alexander Rakowski, Remo Monti, Christoph Lippert.

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
