## [Decision Letter · Decision Letter 0]

29 Jul 2024

Dear Dr Rakowski,

Thank you very much for submitting your Methods entitled 'TransferGWAS of T1-weighted Brain MRI Data from the UK Biobank' to PLOS Genetics.

The manuscript was fully evaluated at the editorial level and by independent peer reviewers. The reviewers appreciated the attention to an important problem, but raised some substantial concerns about the current manuscript. Based on the reviews, we will not be able to accept this version of the manuscript, but we would be willing to review a much-revised version. We cannot, of course, promise publication at that time.

If you decide to revise the manuscript for further consideration at PLOS Genetics, please aim to resubmit within the next 60 days, unless it will take extra time to address the concerns of the reviewers, in which case we would appreciate an expected resubmission date by email to plosgenetics@plos.org.

To resubmit, log into your Editorial Manager account and select the option 'Revise Submission' in the 'Submissions Needing Revision' folder.

We are sorry that we cannot be more positive about your manuscript at this stage. Please do not hesitate to contact us if you have any concerns or questions.

Yours sincerely,

Xiang Zhou, Ph.D.

Academic Editor

PLOS Genetics

Hua Tang

Section Editor

PLOS Genetics

Reviewer's Responses to Questions

**Comments to the Authors:**

Reviewer #1: Rakowski and colleagues used a recently developed approach, i.e., TransferGWAS, to identify more genetic loci associated with neuroimaging-derived phenotypes by extracting low-dimensional representations of imaging data for GWAS. Even though it could be a very sensitive approach to identify more genetic variants that have not been discovered in previous algorithms, there is a subtle improvement of polygenic risk scores in the prediction of phenotypes. The statistical approaches used in the current study sound very robust, but it’s still necessary to rewrite part of the results and clarify how this approach works. I have some comments, which could improve the quality of this study.

1. This study lacks too many details to clarify how the authors investigate the genetics of neuroimaging phenotypes, even though Figure 1 shows the workflow of this study. For example, in the Introduction, the authors mentioned: “We encoded the brain scans using 63 models pre-trained on the ImageNet [48] and Alzheimer’s Disease Neuroimaging Initiative (ADNI) 64 datasets”. However, it’s unclear why they selected two datasets to train the data. Another example is that “As opposed to the ENDO approach…..”. It’s unclear how this ENDO approach works. What’s the difference between ENDO and TransferGWAS. Hopefully, the authors could rewrite the Introduction and introduce more background for this study.

2. In the Introduction, the authors mentioned that “Instead of using manually defined traits, a recent line of work employed deep learning (DL) to derive imaging features using pre-trained deep neural network (DNN)”. Are there any references? What does “a recent line of work” refer to?

3. In Section 3.1, the authors mentioned that “In order to interpret the signal carried by the DNN features, we extracted the first 10 principal components (PCs) of both DNN models”. It’s very confusing. What types of phenotypes did the authors use to extract the top 10 components? When did they extract the top 10, instead of the top 20 components? Since the authors put more Method details in the next section, they should introduce more background about how to confirm the top 10 components.

4. In Section 3.2, the authors tested the heritability of each PC and performed GWAS across all components, which is good. Based on Figure 4, I guess some of the PCs might not have significant heritability. So, it does not make sense to scan genetic variants associated with a trait that is not heritable.

5. I guess the authors performed GWAS for each PC. However, Figure 3 only showed two Manhattan plots for traits derived from ImageNet pretraining and ADNI pretraining. Where are the rest 18 plots? Could the authors explain it?

6. In Section 3.2.3, the authors performed another GWAS using discovery and replication cohorts (23,604 and 12,709 samples). Where do these cohorts come from? Did the authors divide the overall sample into two parts? The authors are missing many details to clarify the results. Hopefully, the authors could add more details to the Results, which could help the readers better understand this study.

7. In Section 3.3, the authors calculate the PGS for all remaining N = 451,450 participants not included in the GWAS sample. Because there are not many details in this section, I am not sure whether this cohort (N=451,450) is a European-ancestry sample. Since GWAS is generated from individuals with European ancestry, the PRS should also be generated in European ancestry samples.

8. Figure 6, I think it would be better if the authors showed these gene-brain associations in a 3D manner. The current 2D visualization could not clearly show these interesting findings, at least I could not see any significant brain regions associated with genetic variants.

9. In the Section 5.2, SNPs were filtered with MAF≥ 0.1%. Since the study aims to investigate common genetic variants associated with brain phenotypes, common genetic variants are typically defined as MAF>1%. I would suggest the authors revise this MAF threshold to avoid false positive signals.

10. In Section 3.3.1, the author performed PheWAS to screen for phenotypes associated with PRS. How did the authors perform phenotypic associations? What type of models did the authors use for testing associations? Given that UKB includes different types of variables (e.g., binary and continuous), do they use a single model for all phenotypes? The authors should add more details to clarify that.

Reviewer #2: In "TransferGWAS of T1-weighted Brain MRI Data from the UK Biobank," the authors provide a deep neural network (DNN) based approach to learn a low dimensional representation of the brain, and then they perform genome-wide association study on this low dimensional representation. Their method TransferGWAS is based on training the DNN (to find the low dimensional representation) on a dataset other than UK Biobank, and then applying the low dimensional representation to UK Biobank, finding 289 associated genetic variants, of which they claim 14 to be novel. They also find a novel link between brain and type 2 diabetes.

This work is relevant to the genetics and neuroscience community. Reliance on preprocessing techniques such as SPM and freesurfer to define volumes of regions of interest is a shortcoming of traditional brain imaging genetics work. Defining brain phenotypes in an unsupervised fashion can improve power, which can lead to new associations, which may improve the clinical relevance of brain imaging GWAS, and further our understanding of the brain. Also, when unsupervised learning is performed on the same dataset that the GWAS is done on, there may be some overfitting leading to false positives. Thus, the idea of TransferGWAS wherein the training stage is done in part on another dataset is interesting.

Major comments:

1. The abstract and introduction of this work however do seem to state quite strongly the extent to which deep neural networks, and transfer learning are novel, and the extent to which voxelwise gwas is not done. The authors cite the ENDO approach of Patel et al. 2022. a) Aside from doing pretraining on a different dataset than the study dataset, in what was is TransferGWAS different from Patel et al. 2022? c) Methodologically, is the application of TransferGWAS in this paper the same as that in Kirchler et al. 2022? c) With regards to voxelwise, there are some pieces from early/mid 2010s such as from Thomas Nichols’ group: “Fast and powerful genome wide association of dense genetic data with high dimensional imaging phenotypes” and Stein et al. “Voxelwise genome-wide association study”. Are there really no updates to voxelwise GWAS since these works? And how does TransferGWAS compare to voxelwise?

2. The authors mention that they train the model for 500 epochs with the Adam optimizer. I understand that this was after pretraining on either ImageNet or ADNI (i.e., these are 500 epochs on UK Biobank. Pretraining on ImageNet has been found to improve accuracy of a resulting classifier, however that improvement can be lost if the length of training on the dataset-specific (target-dataset, in this case UK Biobank) is increased in terms of epochs. The question is this: Is the pre-training improving the nature of the local maximum that the backpropagation on the DNN eventually settles on? Or is it just getting us to the same local maximum (or an equivalent local maximum) faster? Can the authors please investigate this by increasing the number of epochs and plotting a measure of accuracy (or number of positives/amount of overlap of hits/or another unbiased measure of performance) per additional 100 epochs, for a few thousand more epochs? This would be greatly helpful to the community to show if TransferGWAS (or in general, any pretraining method) is truly bringing something that cannot be found when we do not pretrain, but train for longer. What are the compute costs? If 500 epochs takes a day, and if with two weeks of compute we can perform better than pre-training, then there is not much argument for pre-training (two weeks is not a long time).

3. It appears that the downsampling performed by TransferGWAS is quite extensive, downsampling to a side length of 96. Do the authors then introduce a brain mask within the 96x96x96 volume, to select only voxels that intersect the brain? This may considerably reduce the number of voxels that are then fed into the neural net, which could allow less extensive downsampling. I cannot find a mention of this in the paper, regrets if I missed it.

4. Can the authors comment on the clinical or scientific interpretation of the 14 claimed novel variants, and the type 2 diabetes link? I understand that this is not so much a methods paper, as TransferGWAS was already described in Kirchler et al. 2022. What scientific hypotheses regarding the mapping between genotypes, brain phenotypes and disease are supported or refuted by this work?

5. While early work on brain imaging in UK Biobank often focused on the white British subset, the sample size is now large enough that meta-analysis or replication analysis with ancestries other than white British can now be done. How generalizable to ancestries other than white British are the results? An answer to this question is essential, in order to prevent bias from sample inclusion criteria.

6. Please refer to inclusion criteria based on ancestry in a consistent way. On line 332 UKB field 22006 was described “… self-identified as ‘White British’ … ” and on line 9 this set is referred to as “British” (and not specifically white) and on line 179 as “white-British”. I would recommend referring to this population always as “white British”.

Minor comments:

1. Please increase the font size of Figure 5, possibly after rotating it 90 degrees clockwise. Increase font size of Figure 6, after tiling in two columns. Greatly increase font size of Figure 3, possibly tile in one column.

2. Please refer to UK Biobank as “from UK Biobank” rather than “from the UK Biobank”. While “the UK Biobank” is prevalent usage, as UK Biobank is a proper noun, omitting a definite article is preferred.

3. Please make the writing tighter in the introduction and author summary, and avoid cliches such as “At the same time, ” “The growing size, ”

4. In the fourth sentence, the em dash should be longer (its length is currently that of a hyphen, not an em dash).

5. The references need formatting attention. For example, “early alzheimer’s disease” -> “early Alzheimer’s disease” and “Nature genetics” -> “Nature Genetics” and “The relationship between bmi and onset …” -> “The relationship between BMI and onset …” and “PLoS genetics” -> “PLOS Genetics”

6. The tables in the supplementary material could do with captions indicating units, and describing in more detail what the columns mean. I struggled to read the supplementary tables.

Reviewer #3: The authors employ a previously published approach, TransferGWAS (by Kirchler et al.), originally applied to retinal fundus photographs, to a new modality: T1-weighted Brain MRI data from UK Biobank. They find novel genetic findings, and show how these can help in building polygenic scores (PGS) with higher predictive performance for some phenotypes.

Regarding interpretability of the embeddings:

- I believe the "Interpretation of DNN Features" Section should be improved and enlarged. While the presence of many significant correlations with different traits is shown in an aggregated way in Fig. 2 and the individual correlations are included as Supplementary Material, the question of what's the impact of the different embeddings in brain morphology is central and should be addressed. I would suggest distilling the conclusions from the correlation study into a couple of additional paragraphs, so that the reader can better understand what each of the 20 PCs is modelling. An additional figure (e.g. another panel in Fig. 2) showing the correlation of some brain structural traits with the PCs could perhaps help to this end. While some correlations seem to be described in Section 3.4 "Genetic correlations", it's not clear if these are genetic correlations (which the section title suggests) or "direct" correlations (as the text suggests). In any case, I believe Section 3.1 would be a better place to include this information.

- It would also be reassuring to determine whether these insights are confirmed by directly examining the impact of the DNN-derived PC embeddings in the image space. For example, in the case of the VAE network, traversals can be performed on the PCs, e.g. from (mean - 3 sd) to (mean + 3 sd), and examine changes in the output of the decoder. This could help localize the changes to some specific region of the brain. The PCA inverse transformation could be pre-appended to the decoder network as a fully connected layer, so that your DL code can be readily used to create these figures.

In the case of the ImageNet, since you have no decoder you could try to build attribution maps by employing explainable AI techniques. For example, if using Pytorch the library Captum might be handy to this end. The average operation can be added as an additional Pytorch module, and Captum could be used on a 3D model built in this way.

- In Fig. 2, please clarify if you are counting a single association whenever you have more than one significant PC-trait association with a given trait. I understand this is the case since you refer to these values as fractions, however the caption seems to suggest otherwise (i.e. "we plot the number of significant associations per model" this reads as though there could be double-counting). Also, could more fine-grained results be presented, showing whether

Regarding the models:

- Is there an expected benefit in training the VAE in a separate dataset instead of directly on the UK Biobank population? Is this done because the ADNI dataset contains a greater number individuals with brain pathologies which can be used to guide the training? Also related to this, I am curious as to whether the authors have assessed the heritability of features that come from a VAE trained simply with the reconstruction task. The conclusion that a benefit is found in using multi-task learning with a prediction head would be really interesting to report.

- I like the idea of inputting demographic data to the decoder to "explain them away" from the embeddings. The correlation of the embeddings (or the PCs) with age and sex can be reported to ensure that this procedure is having the expected effect.

- I wonder if the data matrix for PCA in each of the two experiments, consists of hidden features that are properly scaled. In principle, the neural network training could converge to a set of hidden features that span different ranges. For instance, hidden feature h1 could span the range [-3, 3] and h2 could span [0.1, 0.2], where the amplitude of these ranges is not necessarily meaningful and which would affect the results of PCA. In particular, some hidden feature with a large range can be overshadowing another, more important one, but with a smaller range. It would be reassuring to know this is taken into account somehow, either by scaling the hidden features to a common range or by justifying why this is not actually desirable.

- The choice of 10 as the number of PCs to be considered should also be justified, e.g. in terms of the proportion of variance explained or the absence of GWAS hits.

- It would be interesting to examine how PCs from the two different models relate to each other. In particular, are they linearly identifiable, i.e. can a linear transformation be applied to one set of PCs to obtain the other? Canonical correlation analysis could be helpful to this end. If the answer is no, i.e. if some PCs can't be accurately reconstructed as linear combinations of the other set of PCs, this would provide evidence that the two networks actually extract orthogonal information, further justifying the use of both.

Some additional minor comments:

- In the abstract, I suggest adding a closing statement regarding the possible implications of the findings.

- For Table 1: is a statistical test being used to declare the improvement significant? If so, what's the test? If not, I would discourage the use of the word "significant" since it normally implies a statistical test being employed, at least in my experience.

- Line 337: Usually, a threshold in INFO imputation score is applied (typically 0.3). Is this filtering step being performed here? It looks to be the case based on the number of SNPs, but it would be good to explicitly mention it.

- For Fig. 3, I suggest adding gene names. Given the large number of loci and the central role of these figures, also consider using landscape orientation and splitting the Manhattan plots into two different figures. It would also be informative to add some indication for the GWAS hits of whether the loci are shared between the two models, or whether effect sizes are significantly higher in one case or the other.

- Line 246: and -> an.

**Have all data underlying the figures and results presented in the manuscript been provided?**

Reviewer #1: Yes

Reviewer #2: **No: **The summary statistics for the GWAS appear to be not available (only summary statistics available for PHEWAS).

Reviewer #3: None

PLOS authors have the option to publish the peer review history of their article (what does this mean?). If published, this will include your full peer review and any attached files.

Reviewer #1: No

Reviewer #2: No

Reviewer #3: No

---

## [Decision Letter · Decision Letter 1]

7 Nov 2024

Dear Dr Rakowski,

We are pleased to inform you that your manuscript entitled "TransferGWAS of T1-weighted Brain MRI Data from UK Biobank" has been editorially accepted for publication in PLOS Genetics. Congratulations!

Yours sincerely,

Xiang Zhou, Ph.D.

Academic Editor

PLOS Genetics

Hua Tang

Section Editor

PLOS Genetics

Aimée Dudley

Editor-in-Chief

PLOS Genetics

Anne Goriely

Editor-in-Chief

PLOS Genetics

Comments from the reviewers (if applicable):

Reviewer's Responses to Questions

**Comments to the Authors:**

Reviewer #1: The authors addressed all of my concerns. I have no comments on this study. I think the current version of the study has improved a lot.

Reviewer #2: The manuscript under review is a revision to "TransferGWAS of T1-weighted Brain MRI Data from UK Biobank." The paper presents a deep learning technique to uncover brain imaging phenotypes. The phenotypes are learned in an unsupervised way, potentially improving on hand-crafted pipelines used to make phenotypes from MRI scans. The manuscript focusses on the transferability of the learned phenotypes. In particular, in the main experiment, the phenotypes are learned on one dataset (Alzheimer’s Disease Neuroimaging Initiative), and then GWAS is performed on another dataset (UKB). This design is crutial to ensure that the phenotypes learned are useful, and not overfitting to the training dataset. This work has the potential to improve brain imaging genetics by reducing the reliance on hand-crafted features.

The authors have made extensive revisions to the initial submission, and improved the manuscript. The revisions include 1) improvements to the clarity, 2) investigation of whether their previous main results were confounded by ancestry, 3) improvements in Figures and clarity of p-values. The author's revisions are all valid, and I think that they cover all of the reviewer's questions. I have no further comments on this manuscript.

Reviewer #3: I thank the authors for properly addressing my comments and those of my fellow reviewers from the previous round of revision.

I believe the manuscript is now in good shape for publication.

I would suggest adding the p-values in Table 1 (or otherwise in the Supplementary Material), since the differences seem too small to be declared significant. This addition would give more reassurance to the readers.

**Have all data underlying the figures and results presented in the manuscript been provided?**

Reviewer #1: Yes

Reviewer #2: Yes

Reviewer #3: Yes

PLOS authors have the option to publish the peer review history of their article (what does this mean?). If published, this will include your full peer review and any attached files.

Reviewer #1: No

Reviewer #2: No

Reviewer #3: No

**Data Deposition**

http://datadryad.org/submit?journalID=pgenetics&manu=PGENETICS-D-24-00628R1

**Press Queries**

---

## [Editor Report · Acceptance letter]

5 Dec 2024

PGENETICS-D-24-00628R1 

TransferGWAS of T1-weighted Brain MRI Data from UK Biobank 

Dear Dr Rakowski, 

We are pleased to inform you that your manuscript entitled "TransferGWAS of T1-weighted Brain MRI Data from UK Biobank" has been formally accepted for publication in PLOS Genetics! Your manuscript is now with our production department and you will be notified of the publication date in due course.

With kind regards,

Anita Estes

PLOS Genetics

On behalf of:
